# Smoking and Neuropsychiatric Disease—Associations and Underlying Mechanisms

**DOI:** 10.3390/ijms22147272

**Published:** 2021-07-06

**Authors:** Omar Hahad, Andreas Daiber, Matthias Michal, Marin Kuntic, Klaus Lieb, Manfred Beutel, Thomas Münzel

**Affiliations:** 1University Medical Center, Department of Cardiology—Cardiology I, Johannes Gutenberg-University Mainz, 55131 Mainz, Germany; omar.hahad@unimedizin-mainz.de (O.H.); marin.kuntic93@gmail.com (M.K.); 2German Center for Cardiovascular Research (DZHK), Partner Site Rhine-Main, 55131 Mainz, Germany; matthias.michal@unimedizin-mainz.de; 3University Medical Center, Department of Psychosomatic Medicine and Psychotherapy, Johannes Gutenberg-University Mainz, 55131 Mainz, Germany; manfred.beutel@unimedizin-mainz.de; 4University Medical Center, Department of Psychiatry and Psychotherapy, Johannes Gutenberg-University Mainz, 55131 Mainz, Germany; klaus.lieb@unimedizin-mainz.de; 5Leibniz Institute for Resilience Research, 55131 Mainz, Germany

**Keywords:** smoking, smoking-induced disease, neuropsychiatric disorder, oxidative stress, inflammation

## Abstract

Despite extensive efforts to combat cigarette smoking/tobacco use, it still remains a leading cause of global morbidity and mortality, killing more than eight million people each year. While tobacco smoking is a major risk factor for non-communicable diseases related to the four main groups—cardiovascular disease, cancer, chronic lung disease, and diabetes—its impact on neuropsychiatric risk is rather elusive. The aim of this review article is to emphasize the importance of smoking as a potential risk factor for neuropsychiatric disease and to identify central pathophysiological mechanisms that may contribute to this relationship. There is strong evidence from epidemiological and experimental studies indicating that smoking may increase the risk of various neuropsychiatric diseases, such as dementia/cognitive decline, schizophrenia/psychosis, depression, anxiety disorder, and suicidal behavior induced by structural and functional alterations of the central nervous system, mainly centered on inflammatory and oxidative stress pathways. From a public health perspective, preventive measures and policies designed to counteract the global epidemic of smoking should necessarily include warnings and actions that address the risk of neuropsychiatric disease.

## 1. Introduction

The tobacco epidemic represents one of the biggest global public health challenges, remaining a leading cause of global morbidity and mortality, despite substantial efforts to combat tobacco use such as the MPOWER measures introduced by the World Health Organization (WHO) in 2008 [1]. The WHO estimates that tobacco use is responsible for more than 8 million global deaths each year, killing up to half of its users and thus making it the single greatest preventable cause of death [2]. While cigarette smoking is a well-established risk factor for non-communicable diseases and related deaths including cardiovascular disease, cancer, chronic lung disease, and diabetes mellitus (Figure 1) [3], much less is known about its impact on neuropsychiatric disease risk as well as the underlying pathomechanisms. Since neuropsychiatric diseases constitute a major contributor the global burden of disease and public health costs, there is an urgent need to identify relevant risk factors and constellations, making smoking a potential target of neuropsychiatric disease prevention strategies. In the present review, we provide an updated overview of the effects of smoking on neuropsychiatric disease risk as well as other main diseases, along with pathophysiological insights from human and animal studies centered on inflammatory and oxidative stress pathways.

## 2. Smoking-Induced Diseases

The latest issue of the Global Burden of Disease study (GBD) from 2019 ranked smoking as the second highest risk factor for attributable disability-adjusted life years (DALYs) [4]. The same study reported that only in Europe 10 to 20% of all DALYs could be attributed to smoking (Figure 2). This finding is not surprising when considering that almost all of the known non-communicable disease groups were at some point correlated with tobacco smoking [5]. Traditionally, smoking was associated with higher incidence of lung cancer and respiratory disease [6], but modern analytical methods and large clinical trials have expanded on these findings [7]. The high amount of toxic compounds generated by the burning of tobacco cigarettes (over 9000 different chemical compounds [8] has been found to play a role in onset and progression of many diseases such as cardio- and neurovascular diseases, metabolic diseases, and neuropsychiatric diseases. Most of the toxic chemical groups found in tobacco smoke are labeled as carcinogenic such as tobacco specific nitrosamines and polycyclic aromatic hydrocarbons [9,10]. Other common chemical groups like volatile organic compounds, that easily pass the blood-brain barrier, and carbonyl compounds, that can create protein adducts, can be implicated in many different diseases beside cancer [11,12]. Carbon monoxide, a product of incomplete burning, is also present in tobacco smoke and is a known human toxicant [13]. Since tobacco smoking is usually not an acute but rather a chronic exposure, accumulated toxic material can affect a whole spectrum of diseases.

### 2.1. Cancer

As far back as 1939, it was noticed that smoking was associated with an increase in lung cancer incidence [14]. It was only in the 1960s, after large epidemiological trials and novel mechanistic insight had emerged, that the general public and the medical and scientific community officially acknowledged the causal link between tobacco smoking and lung cancer [6,15]. Now that the causal link between tobacco smoking and lung cancer is evident, other types of cancer are being not only associated, but also causally linked to tobacco smoking. Breast cancer, the most common type of cancer, was found to be correlated with smoking. A report from the California Environmental Protection Agency from 2005 was the first report of a major health institution, stating that contemporary data provided support for a causal association between current smoking and elevated breast cancer risk [16]. In the following years, other institutions have provided reports, indicating at least a correlation between tobacco smoking and incidence of breast cancer [17,18]. A recent meta-analysis found that women who smoked during their life had an overall relative risk (RR) of 1.10 in prospective studies and 1.08 in retrospective studies of developing breast cancer [19]. The overall RR increased to 1.13 for current smokers in prospective studies.

Prostate cancer, the third most common type of cancer, is not directly correlated with tobacco smoking [20], but tobacco smokers have poorer survival and recurrence-free rates after prostate cancer diagnosis [21]. Bladder cancer, on the other hand, is known to be caused by tobacco smoking [20]. In a meta-analysis including 107 studies on bladder cancer, the RR for all smokers was 2.58 and even reached 3.47 for current smokers [22].

Other types of cancer were also correlated with tobacco smoking. Increased risk of cervical cancer was correlated with smoking, with an odds ratio (OR) of 2.17 for human papillomavirus positive women [23]. The risk of developing cervical cancer was also found to be elevated in passive smokers, indicating that even women who do not actively smoke are at risk [24]. Pancreatic cancer was also observed to be correlated with tobacco smoking. A recent meta-analysis reported an RR of 1.66 when compared to never smokers [25]. As with prostate cancer, the survival after a pancreatic cancer diagnosis was reduced with a hazard ratio (HR) of 1.37 compared to never smokers and the HR increased with the amount of tobacco consumed (pack-years) [26]. Tobacco smoking is also found to be linked with oral cancers [27], head and neck cancers [28], and ovarian cancer [29].

### 2.2. Lung Disease

Studies from the 1960s recognized that smoking causes not only lung cancer, but other types of respiratory disease such as chronic bronchitis [15]. Chronic obstructive pulmonary disease (COPD) is one of the most common lung diseases associated with tobacco smoking. It is now accepted that tobacco smokers have a much higher risk of developing COPD and that almost 50% of all smokers are expected to develop COPD during their life [30]. Smoking cessation after COPD diagnosis is the major factor for reducing mortality [31]. Smoking was also associated with higher incidence of adult asthma in Finland, with an OR of 1.33 [32]. Globally, it is not clear if there is a causal link between smoking and asthma, but it is clear that smoking in asthmatic populations leads to higher risk of developing more severe pulmonary disease like COPD [33]. Interstitial lung diseases that result in damage of lung interstitial tissue were shown to be smoking-dependent [34]. Pulmonary Langerhans cell histiocytosis is an example of a pulmonary disease that is almost completely dependent on smoking, as smoking is considered a major, and maybe the only, risk factor [35]. Respiratory bronchiolitis is an inflammatory response to chronic tobacco smoke injury and is only curable by smoking cessation [36]. From all of the cases of adult desquamative interstitial pneumonia, over 90% are smokers [37]. Smoking is also a recognized risk factor for idiopathic pulmonary fibrosis, however has not been causally linked [38].

### 2.3. Cardiovascular Disease

Cardiovascular disease is the most common cause of death worldwide [39]. Proper function of the endothelium is of the essence for a healthy cardiovascular system. When the endothelium is exposed to oxidative stress or inflammatory mediators, it can become injured and dysfunctional. Tobacco smoke is shown to have an influence on endothelial function, as many studies have demonstrated that smokers have reduced flow-mediated dilation in comparison to non-smokers [40,41], but also reduced endothelium-dependent dilation in response to intrabrachial and intracoronary acetylcholine infusion [42,43]. Endothelial dysfunction can be attributed to the direct toxic effects of chemical components and reactive oxygen species (ROS) being part of the tobacco smoke [41], including superoxide and peroxynitrite [42], a highly reactive intermediate generated by the reaction between superoxide and nitric oxide (NO) [44], and many pro-inflammatory chemicals that can accelerate the progression of atherosclerosis [45] (for summary, see Figure 3).

The effects of smoking on blood pressure are not as clear as the effects on endothelial function. Nicotine is an alkaloid that is known to acutely increase blood pressure and heart rate, but can also cause an increase in flow-mediated dilation due to the increased cardiac output [46]. It would follow that prolonged exposure to nicotine could induce hypertension, but the causal connection is unclear. In general, it is accepted that smoking cessation improves arterial stiffness and reduces risk of further cardiovascular disease in people with hypertension [47].

Tobacco smoking shows a strong association with other cardiovascular diseases like coronary artery disease, heart failure, and atrial fibrosis. A recent meta-analysis has shown that smoking only one cigarette per day carries 50% of the risk of developing coronary artery disease as smoking 20 cigarettes per day [48]. Smoking cessation was also shown to reduce the risk of developing coronary artery disease in a time-dependent manner [49]. Chronic increase of blood pressure in smokers is a potent risk factor for heart failure. The inhaled carbon-monoxide can lead to chronic ischemia and contribute to the development of heart failure. Interestingly, the prolonged ischemia could lead to ischemic preconditioning and may be the reason why smokers have better outcomes after myocardial infarction, also known as the “smokers paradox” [50]. Increased risk of atrial fibrillation was found to be associated with smoking, with an RR of 1.51 and 1.49 for current and former smokers, respectively [51].

### 2.4. Metabolic Disease

Tobacco smoking has been observed to increase the risk of developing type 2 diabetes mellitus and metabolic disease in general. In a cohort questionnaire study of 40,000 men, smokers had an increased risk of developing diabetes, with an OR of 1.94 in comparison to non-smokers [52]. Other studies have found a more moderate risk increase, with 1.25 and 1.28 for medium and heavy smokers [53]. In a large European study, non-diabetic increase in blood glucose was also associated with smoking [54]. The authors observed that the increase in smoking frequency (pack-years) is correlated with the increase in plasma hemoglobin A1c, a marker of blood glucose level. Just like with other diseases, cessation of smoking results in a decreased risk of developing diabetes. A study following postmenopausal women showed that after 10 years of cessation, the risk of developing diabetes was the same as in never smokers [55]. Of note, Hahad et al. recently demonstrated that microvascular endothelial dysfunction is strongly predictive of incident prediabetes and type 2 diabetes mellitus in the general population, providing a mechanistic basis by which smoking (and other associated risk factors) does not only increase the risk of cardiovascular disease, but also the development and progression of type 2 diabetes mellitus [56].

## 3. Smoking-Induced Disease Mechanisms

### 3.1. Oxidative Stress and Inflammation

The presence of such a wide range of different toxic chemicals in tobacco smoke can explain the diverse group of diseases that are associated with smoking. On the mechanistic level, smoking exerts its toxicity and contributes to disease onset and progression by two major mechanisms. Mainly oxidative stress, both from free radicals in smoke and from oxidative imbalance in the cells by secondary activation of intrinsic sources of ROS, and inflammation are responsible for the majority of the long-term negative effects of smoking. Here, we will briefly cover these basic mechanisms.

Oxidative stress is an imbalance in the activation of pro-oxidative and antioxidant pathways that results in an increased concentration of ROS (for summary see Figure 3). Because the tobacco is burned in an oxygen-rich environment, smoke contains many organic free radicals and ROS [57]. Most notably, superoxide radical can react with NO from endothelial cells to create peroxynitrite, which reduces the bioavailability of NO [58]. Reduction in NO availability is then responsible for impaired endothelial signaling and endothelial dysfunction. Peroxynitrite can easily nitrate protein tyrosine residues, causing additional oxidative damage in the cells [59]. Other free radicals present in tobacco smoke together with peroxynitrite can cause lipid peroxidation, resulting in oxidation of low-density lipoprotein (LDL) [60]. Oxidized LDL is known to be implicated in the development and progression of metabolic syndrome [61]. It is not only the free radicals in the smoke that cause oxidative damage in the cells, as other reactive chemicals such as aldehydes can induce activation of pro-oxidative enzymes. Acrolein from cigarette smoke can cause activation of nicotinamide adenine dinucleotide phosphate (NADPH) oxidase, the enzyme responsible for production of superoxide radicals in phagocytic but also vascular and other cells [62]. Increased oxidative stress is also detrimental to the mitochondria of endothelial cells as they are very susceptible to marginal changes in free radical concentrations. Smoking-induced mitochondrial dysfunction can further lead to endothelial dysfunction and progression of cardiovascular disease [63].

Prolonged oxidative stress together with noxious chemicals from smoke can also lead to inflammation, a major contributor to cardiovascular disease [64,65]. Smoking status was associated with higher levels of high-sensitivity C-reactive protein (a marker of inflammation) in blood [66] as well as with intercellular adhesion molecule and interleukin (IL)-6 [67]. Smoking promotes the migration and accumulation of neutrophils and macrophages in the lung, but also desensitizes them to foreign pathogens, making smokers more susceptible to infections [68]. Alveolar macrophages serve as a first line of defense against airborne pollutants, and smoking increases their number and activation status [69]. Activated macrophages release ROS and can cause tissue damage, which links smoking to a variety of lung diseases. Macrophages also release pro-inflammatory cytokines that are distributed by blood and cause systemic inflammation [67]. Systemic inflammation is a known risk factor for atherosclerosis and other cardiovascular diseases [70]. Another link between oxidative stress and inflammation is the ability of nicotine to induce IL-8 secretion in neutrophils, which is mediated through peroxynitrite activation of nuclear factor kappa-light-chain-enhancer of activated B cells (NF-κB) [71]. Platelet aggregation was found to be elevated in smokers [72], indicating that blood clotting and risk of stroke and myocardial infarction could be elevated as well.

### 3.2. Implications for Neuropsychiatric Disease

The ability of tobacco smoke to cause oxidative stress and systemic inflammation has far-reaching consequences. Besides causing cancer, lung disease, and cardiovascular disease, tobacco smoking is associated with different neuropsychiatric diseases.

Lower activity of the antioxidant enzyme paraoxonase was associated with major depression disorder [73]. The same study found that smoking reduces the activity of paraoxonase, which could classify smoking as a risk factor for major depression disorder acting through oxidative stress. Presence of 8-hydroxy-2-deoxyguanosine, a marker of DNA damage by free radicals, was found in the urine and plasma of patients suffering from depression [74]. Increased NO protein modifications were observed in the serum of depressed patients, further strengthening the link between nitrosative stress and depression [75]. Parental smoking is also a risk factor for developing autism spectrum disorder in the offspring, based on the induced oxidative stress [76]. Both oxidative stress and inflammation were increased in the brains of post-traumatic stress disorder model animals [77]. Oxidative stress markers are prominent features of neurodegenerative diseases and neuropsychiatric disorders. Oxidative stress leads to lipid peroxidation and the creation of products such as 4-hydroxynonenal. 4-hydroxynonenal can induce inflammation and apoptosis of neuronal cells, lead to accumulation of peroxides in astrocytes [78], impairment in axon regeneration, aberrations in axonal functioning, loss of active mitochondria, and suppression of mitochondrial respiration [79,80]. All of these detrimental effects of oxidative stress have a high impact on neurodegenerative diseases like Parkinson’s disease and Alzheimer’s disease. Loss of motor function in Parkinson’s disease caused by degradation of dopaminergic neurons is promoted by an accumulation of alpha-synuclein. Accumulation of alpha-synuclein is both increased and increases oxidative stress itself, producing a neurodegeneration cycle [81]. Compelling data demonstrates that impaired neuronal metal homeostasis could be involved in the formation of oxidative stress, influencing amyloid aggregation in case of Alzheimer’s disease [82].

High levels of cytokines have been shown to associate with anxiety and depressive mood [83], leading to a hypothesis that chronic inflammation could be responsible for these psychiatric diseases [84]. Higher levels of circulating cytokines, mostly IL-6 and tumor necrosis factor alpha (TNFα), were found in patients with schizophrenia and bipolar disorder [85]. Microglia activation observed through translocator protein binding, a measurement of neuroinflammation, demonstrated that inflammation is implicated in obsessive-compulsive disorder [86]. Attention-deficit/hyperactivity disorder (ADHD) is a neuropsychiatric disease that affects children, but it was observed that parental smoking was a risk factor for developing ADHD [87]. This association was made through the presence of pro-inflammatory cytokines in ADHD children whose fathers were smokers [88]. Same as with oxidative stress, inflammation is implicated in neurodegenerative disease as well, leading to neuropsychiatric features. Both Parkinson’s and Alzheimer’s disease are accompanied by local and systemic inflammation [89]. The relationship between smoking, oxidative stress, inflammation, and neuropsychiatric disease is not always clear. This stems from the fact that many neuropsychiatric diseases also increase the chance that a person will start smoking [90], making the direction of association difficult to establish (Figure 4). The clear distinction between cause and effect is a major issue in previous studies.

Tobacco smoking is not only implicated in neuropsychiatric disease through oxidative stress and inflammation, but also through direct exposure to some of the chemicals present in tobacco smoke. High environmental concentrations of lead have caused its accumulation in plants and the tobacco plant is no exception. Lead is known to induce schizophrenia [91] and increased intake of lead from tobacco is a risk factor for developing schizophrenia later in life. Nicotine, a highly addictive substance that modifies neurotransmitter patterns, can have a direct influence on neuropsychiatric diseases like schizophrenia and depression [92]. Interestingly, there is also evidence for procognitive effects of nicotine in subjects with neuropsychiatric diseases such as schizophrenia [93,94] and ADHD [95,96], which may explain sustained smoking in these disease phenotypes. In addition, chemicals present in smoke may interact with antipsychotics, antidepressants, and benzodiazepines through pharmacokinetic and pharmacodynamic (mainly nicotine-mediated) pathways [97], showing that, for example, smoking cessation in patients receiving clozapine may lead to elevated plasma concentrations of clozapine and severe side effects via altered metabolic clearance by CYP1A2 [98]. More details on the contribution of nicotine to neuropsychiatric disorders were reviewed in references [92,99,100]. Lastly, it is highly important to note that (neuro)psychiatric disorders have a strong link with chronic stress, which represents one of the more prominent risk factors for their onset. In this context, cancer, cardiovascular, and metabolic disorders, together with many other chronic pathologies, represent severe forms of chronic stress, intuitively increasing the risk of neuropsychiatric disorders.

## 4. Smoking and Neuropsychiatric Disease Risk

Although there is ample evidence indicating an interrelationship between smoking and neuropsychiatric diseases, meaning that rates of smoking are markedly higher in subjects with prevalent neuropsychiatric disease than in the general population being two to five times higher including subjects with, for example, schizophrenia, depression, anxiety disorders, ADHD, binge eating disorders, bulimia, and substance use disorders [102] (Figure 4), much less is known about the prospective impact of smoking on neuropsychiatric disease development. However, emerging strong evidence from epidemiological studies suggests that smoking may be a causal factor for the development or progression of neuropsychiatric disease.

### 4.1. Evidence from Observational/Epidemiological Studies

#### 4.1.1. Depression

In the Gutenberg Health Study, a prospective cohort study from Germany, Beutel et al. evaluated longitudinal data of 10,036 participants, demonstrating that current smoking was predictive of new onset of depression, with an OR of 1.35 (95% confidence interval (CI) 1.05–1.71), which remained stable after further adjustment for subclinical depression at baseline [103]. Cabello et al. analyzed data from 7908 participants from Ghana, India, Mexico, and Russia from the WHO’s Study on Global Ageing and Adult Health, showing that current daily (OR 1.46, 95% CI 1.09–1.97) as well as non-daily (OR 2.06, 95% CI 1.18–3.62) smoking was associated with incident depression [104]. In the Copenhagen City Heart Study (N = 18,146), smoking more than 20 g of tobacco per day was independently associated with incident depression in women (HR 2.17, 95% CI 1.45–3.26) and men (HR 1.90, 95% CI 1.05–3.44) after a follow-up of up to 26 years [105]. A Norwegian study (N = 1190) analyzed the association between smoking and subsequent first depression [106]. The authors demonstrated that HRs for incident depression followed a dose-dependent association for former and current smoking, with current smokers who smoked more than 20 cigarettes per day having the highest risk of developing depression (HR 4.34, 95% CI 1.85–10.18). In an Australian case-control study, smoking was cross-sectionally (N = 165 cases and 806 controls, age-adjusted OR for smoking more than 20 cigarettes per day 2.18, 95% CI 1.31–3.65) and prospectively (N = 671, HR 1.93, 95% CI 1.02–3.69, not explained by physical activity or alcohol consumption) associated with increased risk of major depressive disorder in women [107]. Interestingly, on the basis of the National Longitudinal Study of Adolescent Health from the United States (US), Goodman and Capitman revealed that current smoking in adolescents (N = 8704) was strongly predictive of developing high depressive symptoms, whereas in non-current smoking adolescents (N = 6947) high depressive symptoms were not predictive of heavy smoking after multivariable adjustment, challenging the common assumption of causality in the context of smoking and depression that rather goes in the direction from depression to smoking [108]. In good agreement, in the Korea Welfare Panel, smoking was shown to predict depression, whereas no association was found when testing for the opposite direction for the relationship of depression and smoking [109]. Conversely, Munafò et al. showed within the same cohort that baseline smoking status did not predict depressed mood at follow-up in adolescents, although there was a trend towards a significant relationship in females [110]. A more recent systematic review on the association between smoking, depression, and anxiety including a total of 148 studies found that smoking status was positively associated with later depression in most studies (37 out of 51 studies), while relatively few studies (14 out of 51 studies) found no evidence for this relationship [111]. Evidence for a bidirectional relationship between smoking and poor mental health/depressive mood arises from a longitudinal analysis of young Australian women (N = 10,012), indicating that smoking was associated with higher odds of depressive mood at subsequent waves, with smoking more than 20 cigarettes per day displaying the highest odds [112]. Conversely, women with poor mental health/depressive mood had higher odds of smoking at subsequent waves. Khaled et al. analyzed data from the Canadian National Population Health Survey, demonstrating that heavy smoking was associated with onset of major depression (HR 3.1, 95% CI 1.9–5.2) even after adjustment for mental stress [113].

#### 4.1.2. Anxiety Disorder

Using data from the Children in the Community Study from the US (N = 688), Johnson et al. provided evidence that heavy cigarette smoking (≥20 cigarettes/day) during adolescence was associated with elevated risk of agoraphobia (OR 6.79, 95% CI 1.53–30.17), generalized anxiety disorder (OR 5.53, 95% CI 1.84–16.66), and panic disorder (OR 15.58, 95% CI 2.31–105.14) during early adulthood after comprehensive adjustment for confounders, while no evidence was found for the opposite direction of the relationship [114]. Among 34,653 participants from the US National Epidemiologic Survey on Alcohol and Related Conditions, smoking was related to new onset of anxiety disorders including generalized anxiety disorder, panic disorder, social anxiety disorder, specific phobias, and posttraumatic stress disorder, with participants smoking a larger number of cigarettes per day displaying an increased trend of new-onset disorders [115]. Cuijpers et al. used data from the Netherlands Mental Health Survey and Incidence Study (N = 7076) to show that smoking was associated with first-ever incidence of mental disorders, including any anxiety disorder (incidence rate ratio (IRR) 1.77, 95% CI 1.10–2.86), generalized anxiety disorder (IRR 3.80, 95% CI 1.09–13.21), and agoraphobia (IRR 4.07, 95% CI 0.88–18.85) after multivariable adjustment [116]. Likewise, with data from the Young in Norway Longitudinal Study (N = 1501), a 13-year longitudinal study, revealed that smoking was associated with increased risk of mental health conditions including anxiety, whereas measures of impaired mental health were not predictive of later smoking initiation [117]. This was also the case in an Indian study of 2494 women [118]. Conversely, Taylor et al. examined the potential causal relationship between smoking, depression, and anxiety using Mendelian randomization meta-analysis including 25 studies (N = 127,632), revealing no substantial association between the minor allele of rs16969968/rs1051730 (genetic variant as a proxy for smoking heaviness) in current smokers and anxiety (OR 1.02, 95% CI 0.97–1.07) as well as depression (OR 1.00, 95% CI 0.95–1.05) or psychological distress (OR 1.02, 95% CI 0.98–1.06) [119].

#### 4.1.3. Suicidal Behavior

A meta-analysis including 63 studies with 8063,634 participants revealed current smoking to be associated with elevated risk of suicidal ideation (OR 2.05, 95% CI 1.53–2.58), suicide plan (OR 2.36, 95% CI 1.69–3.02), suicide attempt (OR 2.84, 95% CI 1.49–4.19), and suicide death (RR 1.83, 95% CI 1.64–2.02) [120]. Likewise, a meta-analysis of 15 prospective cohort studies including 2395 cases among 1,369,807 participants suggested that former smoking increased the risk of completed suicide by 28% (RR 1.28, 95% CI 1.001–1.641) and current smoking by 81% (RR 1.81, 95% CI 1.50–2.19) [121]. In a prospective US study of 1200 subjects, Breslau et al. found current daily smoking to increase the risk of suicidal thoughts or attempt by 82% (OR 1.82, 95% CI 1.22–2.69) after adjustment for prior depression and substance use disorders, which remained stable when controlling further for suicidal predisposition, indicated by prior suicidality, and prior psychiatric disorders [122]. Results from three prospective cohorts (Health Professionals Follow-up Study, Nurses’ Health Study I and II) of US men and women (N = 253,033) displayed an increased risk of completed suicide in current smokers (RR 2.69, 95% CI 2.11–3.42) as well as in former smokers (RR 1.15, 95% CI 0.91–1.45), with those smoking more than 25 cigarettes per day experiencing the highest risk (RR 4.13, 95% CI 2.96–5.78) [123]. In a large twin cohort (N = 16,282 twin pairs) with a 35-year follow-up from Finland, smokers had an increased risk of suicide (HR 2.56, 95% CI 1.43–4.59) after excluding those who developed serious somatic or psychiatric illness and after adjustment for depressive symptoms, alcohol, and sedative–hypnotic use [124]. In twin pairs, discordant for smoking and suicide, odds of suicide were elevated in smokers (OR 6.0, 95% CI 2.06–23.8). Conversely, in the Finnish 5-year prospective Vantaa Depression Study of psychiatric patients (N = 269) with major depressive disorder, smoking did not predict suicidal ideation or suicide attempts [125]. This was also the case in a cohort of adolescents (N = 764) in which smoking was not predictive of subsequent suicidal ideation after adjustment for high levels of stress and depression as well as low levels of parental attachment [126]. Prospective data from the German Early Developmental Stages of Psychopathology study were used to demonstrate that prior occasional regular smoking and nicotine dependence were associated with elevated risk for new onset of suicide ideation (OR range from 1.5–2.7) in adolescents and young adults, which remained stable after the exclusion of participants with major depression [127].

#### 4.1.4. Dementia/Cognitive Decline

Strong evidence for an association between smoking and dementia/cognitive decline arises particularly from multiple meta-analyses of prospective studies. In the meta-analysis from Anstey et al. including 19 prospective studies on the association of smoking with dementia (N = 26,374) and cognitive decline (N = 17,023) in the elderly, current smoking increased the risk of incident Alzheimer’s disease by 79% (RR 1.79, 95% CI 1.43–2.23), by 78% for incident vascular dementia (RR 1.78, 95% CI 1.28–2.47), and by 27% for any dementia (RR 1.27, 95% CI 1.02–1.60) [128]. This was accompanied by greater yearly cognitive declines at follow-ups (β −0.13, 95% CI −0.18–−0.08) in current smokers. In good agreement, a more recent meta-analysis from Zhong et al. including 37 prospective studies showed an increased risk of all-cause dementia (RR 1.30, 95% CI 1.18–1.45), Alzheimer’s disease (RR 1.40, 95% CI 1.13–1.73), and vascular dementia (RR 1.38, 95% CI 1.15–1.66) in smokers [129]. A further meta-analysis of 31 studies investigating a range of determinants of incident Alzheimer’s disease provided an RR of 1.37 (95% CI 1.23–1.52) for current/ever smokers [130]. Furthermore, the estimated population attributable risk percent was in particular increased for smoking (31.09%, 95% CI 17.9–44.3). The most recent meta-analysis from Li et al. including 34 prospective studies found current smoking (RR 1.61, 95% CI 1.32–1.95), among other risk factors such as obesity, diabetes, hypertension, and hypercholesterolemia (all of which are interrelated to smoking), to be associated with dementia [131]. In a recent study from Japan (N = 12,489), risk of incident dementia in response to smoking as well as to years since smoking cessation was evaluated [132]. A 46% (HR 1.46, 95% CI 1.17–1.80) higher risk of dementia in response to current smoking was found and, interestingly, abstinence from smoking for at least 3–5 years resulted in a dementia risk that was comparable to those who have never smoked (HR 1.03, 0.70–1.53). In good accordance, in the prospective Korean National Health Screening Cohort (N = 46,140 men), long-term quitters (4 years or more abstinence from smoking, HR 0.86, 95% CI 0.75–0.99) and never smokers had decreased risk of overall dementia (HR 0.81, 95% CI 0.71–0.91) [133]. Moreover, never smoking was associated with lower risk of Alzheimer’s disease (HR 0.82, 95% CI 0.70–0.96) and long-term quitting (HR 0.68, 95% CI 0.48–0.96), and never smoking (HR 0.71, 95% CI 0.54–0.95) was associated with decreased risk of vascular dementia. In contrast, a meta-analysis including data from seven low- and middle-income countries (N = 11,143) indicated no increased risk of Alzheimer’s disease or vascular dementia when analyzing smoking status or cumulative smoking exposure (pack-years) [134].

#### 4.1.5. Schizophrenia/Psychosis

The meta-analysis of Gurillo et al. reported (after including a total of 61 studies) an RR of 2.18 (95% CI 1.23–3.85) for new psychotic disorders in daily smokers [135]. Moreover, onset of psychotic illness was at earlier age in daily smokers compared to non-smokers (−1.04 years, 95% CI −1.82–−0.26). In a more recent meta-analysis from Hunter et al. including 12 studies, smokers had a two-fold increased risk of schizophrenia (RR 1.99, 95% CI 1.10–3.61) [136]. Importantly, prenatal smoke exposure increased the risk of schizophrenia substantially (RR 1.29, 95% CI 1.10–1.51). In a 15-year follow-up cohort from Finland (N = 6801), smoking more than 10 cigarettes per day was associated with increased risk of subsequent psychosis after adjustment for baseline psychotic experiences (HR 2.87, 95% CI 1.76–4.68), remaining stable even after further control for well-established risk factors such as cannabis use, frequent alcohol use, other illicit substance use, parental substance abuse, and psychosis [137]. In a large cohort of Swedish women (N = 1,413,849) and men (N = 233,879), light (HR 2.21, 95% CI 1.90–2.56 for women and HR 2.15, 95% CI 1.25–3.44 for men) and heavy smokers (HR 3.45, 95% CI 2.95–4.03 for women and HR 3.80, 95% CI 1.19–6.60 for men) were at increased risk of first-onset of schizophrenia [138]. A Mendelian randomization study using data from the UK Biobank (N = 462,690) provided evidence for a potential causal link between smoking and the risk of schizophrenia (OR, 2.27, 95% CI 1.67–3.08) as well as depression (OR 1.99, 95% CI 1.71–2.32) [139].

#### 4.1.6. Other Neuropsychiatric Outcomes

In the Nurses’ Health Study II (N = 116,363 women), risk of incident seizure or epilepsy in response to smoking, caffeine use, and alcohol intake was examined [140]. Only smoking was shown to be associated with increased risk of incident events, with current smoking displaying an RR of 2.60 (95% CI 1.53–4.42) for seizure and an RR of 1.46 (95% CI 1.01–2.12) for epilepsy in response to former smoking.

Surprisingly, there is consistent evidence from large prospective studies and meta-analyses that smoking may decrease the risk of Parkinson disease. Most recently, a 65-year follow-up of 30,000 male British physicians from the British Doctors cohort study showed that baseline smoking may reduce risk of Parkinson disease by almost 30% (RR 0.71, 95% CI 0.60–0.84) and continued smoking was determined by resurveys to perhaps reduce risk by 40% (RR 0.60, 95% CI 0.46–0.77) [141]. These results were also confirmed in a European cohort including eight countries [142]. Recent meta-analytic evidence suggested that even secondhand smoke had protective effects in terms of Parkinson’s disease onset, irrespective of exposure occasions and timing (at home: RR 0.73, 95% CI 0.56–0.95, at work: RR 0.82, 95% CI 0.57–1.17, in children: RR 0.91, 95% CI 0.76–1.08) [143].

A meta-analysis of 15 studies on maternal smoking during pregnancy and risk of autism spectrum disorder in offspring found no evidence for an association (OR 1.02, 95% CI 0.93–1.12) [144]. Comparable results were achieved by a further meta-analysis including 22 studies (OR 1.16, 95% CI 0.97–1.40), while a substantial relationship was revealed after considering population-level smoking metrics (i.e., adult male smoking prevalence) [145]. In support of this, maternal smoking during the whole pregnancy was associated with an increased risk of pervasive developmental disorder in the offspring in a Finnish cohort study (OR 1.2, 95% CI 1.0–1.5), which remained stable after further adjustment for maternal age, mothers socioeconomic and psychiatric status, and infants weight for gestational age [146]. In contrast, Caramaschi et al. found no compelling evidence for a causal association between maternal smoking during pregnancy and offspring autism or related traits using Mendelian randomization and a genetic variant at the CHRNA3 locus in maternal DNA as a proxy for heaviness of smoking [147].

Meta-analytic evidence including 20 studies indicated that smoking during pregnancy may lead to higher risk of offspring ADHD (OR 1.60, 95% CI 1.45–1.76), with greater increases in children whose mothers were heavy smokers (OR 1.75, 95% CI 1.51–2.02) compared to light smoking mothers (OR 1.54, 95% CI 1.40–1.70) [148]. Likewise, the meta-analysis of 27 studies yielded results that showed that either prenatal exposure to maternal smoking during pregnancy or smoking cessation during first trimester was related to childhood ADHD after adjusting for parental psychiatric history and social socioeconomic status, while smoking cessation before pregnancy was not associated with childhood ADHD [149]. This was confirmed in a further meta-analysis of 12 cohort studies including 17,304 pregnant women, suggesting that maternal smoking during pregnancy may increase the risk of ADHD by 58% (RR 1.58, 95% CI 1.33–1.88) [150]. A Finnish nested case-control study (N = 10,132 cases and 38,811 controls) demonstrated that maternal smoking during pregnancy may increase the risk of ADHD (OR 1.75, CI 95% 1.65–1.86) [151]. Compared to ADHD cases without comorbidities, a higher risk estimate was observed for subjects with comorbid conduct disorder or oppositional defiant disorder (OR 1.80, CI 95% 1.55–2.11).

### 4.2. Smoking and Brain Changes Affecting Neuropsychiatric Pathophysiology

The link between oxidative stress and neuronal damage is evident from numerous reports. Some of the major toxic compounds in cigarette smoke are especially well known to cause systemic and cardiovascular oxidative stress (Figure 3), which also holds true for the brain, as reported for particulate matter [152], heavy/transition metal ions [153,154,155], reactive aldehydes [156,157,158], and volatile organic compounds [159]. The following animal experimental studies support oxidative stress-mediated neuronal damage that may contribute to the aforementioned neuropsychiatric disease in smokers. Rosiglitazone, by induction of PPARγ and Nrf2 activity, protects against cigarette smoke-induced loss of blood-brain-barrier integrity via reduced inflammation and oxidative stress [160]. Chronic cigarette smoke exposure drives spiral ganglion neuron loss in mice [161]. The neurotoxic effects of smoking were further supported by cerebral aneurysm development and ROS formation by cigarette smoke extract and a central role of the phagocytic NADPH oxidase in this process was demonstrated by an improvement of the adverse phenotype by genetic *p47phox* deletion [162]. Maternal cigarette smoke exposure also enhanced brain inflammation and oxidative stress in male mice offspring [163]. Oxidative stress induced by cigarette smoke in Lewis rat brains was evident by increased brain gene expression for the pro-oxidants *iNOS* and the NADPH components *NOX4*, *dual oxidase 1*, and *p22phox* as well as decreased levels of serum levels of glutathione and, in the brain, similar levels of superoxide dismutase and decreased thioredoxin [164]. Cigarette smoke exposure also induced various types of oxidative DNA damage, including single-strand breaks, double-strand breaks, and DNA-protein cross-links in the brain of neonatal mice, all of which were associated with the oxidative stress marker malondialdehyde and represent important risk factors for both neurodevelopmental and neurodegenerative disorders [165]. Accordingly, moderate exercise prevented cigarette smoke exposure-induced hippocampal oxidative stress (malondialdehyde and protein carbonyls) and adverse neurological behaviors in mice [166]. In cultured cells, these adverse effects were confirmed by higher cellular ROS formation and induction of the stress-response by NRF-2, as well as by impaired blood-brain-barrier function. In vivo, the adverse effects on stroke damage were normalized by metformin treatment, which is known to confer antioxidant protection by AMPK activation [167].

The link of inflammation to neuronal damage is evident from numerous reports. Some of the major toxic compounds in cigarette smoke are especially well known to cause systemic and cardiovascular inflammation (Figure 3), which also holds true for the brain, as reported for particulate matter [152,168], heavy/transition metal ions [169,170], reactive aldehydes [171,172], and volatile organic compounds [173,174]. The following experimental animal studies support inflammation-mediated neuronal damage that may contribute to aforementioned neuropsychiatric disease in smokers. Cognitive spatial tests showed that the mice exposed to cigarette smoke had delayed time in finding food rewards, which was associated with inflammatory responses such as necrosis and cytoplasm vacuolization, as well as higher TNFα expression [175]. Astaxanthin protects the brain against neuroinflammation (normalization of IL-6 and TNFα levels), synaptic plasticity impairment, and oxidative stress (normalization of malondialdehyde) in the cortex and hippocampus of cigarette smoke exposed mice [176]. Third-hand smoke exposure of mice for 4 weeks increased levels of circulating inflammatory cytokines, TNFα, and granulocyte macrophage colony-stimulating factor, as well as the stress hormone epinephrine in the serum and brain tissue [177]. Exposure of mice to tobacco cigarette smoke resulted in a clear increase in markers of inflammation (ICAM-1, PECAM-1, VCAM-1) and ischemic brain damage after stroke [167]. Inflammation induced by cigarette smoke in Lewis rat brains was evident by increased *IFN-γ, TNF-α, IL-1α, IL-1β, IL-23, IL-6, IL-23, IL-17, IL-10, TGF-β, T-bet*, and *FoxP3* gene expression, including upregulation of other central inflammatory mediators *MIP-1α/CCL3*, and less prominent *MCP-1/CCL2* [164]. Exposure to cigarette smoke enhanced post-ischemic brain injury, inflammation by mobilization of neutrophils and monocytes, whereas inhibition of NLRP3 inflammasome prevented these adverse effects [178].

Air pollution constituents such as particulate matter, that is also present at high concentrations in cigarette smoke, may contribute to the onset and progression of dementia (e.g., Alzheimer’s disease) as shown by numerous animal studies (reviewed in [179]). Moreover, exposure to cigarette smoke exacerbated amyloid pathology in a mouse model of Alzheimer’s disease [180]. In addition, a Parkinson’s disease-like phenotype and cognitive deficits such as memory impairment was reported for particulate matter-exposed animals (reviewed in [181]). The observed neuropsychiatric disorders are potentially based on impaired neurotransmitter signaling, upregulation of cerebral cytokines, activation of neuronal immune cells and disruption of the blood-brain barrier, as well as induction of oxidative stress (e.g., as indicated by higher levels of oxidized LDL) (reviewed in [152]). These are pathomechanisms that are also potentially triggered by cigarette smoke [182]. Most likely, these processes facilitate the onset and progression of mental and psychiatric disorders (reviewed in [183,184]), where neuroinflammation and cerebral oxidative stress represent central pathomechanisms triggered by the particles or reactive gases (reviewed in [185,186,187,188]) that are also present in cigarette smoke. Dysregulated microglia, through the releasing neurotoxic cytokines (e.g., TNFα, IL-1β, and INF-γ), as well as different ROS (e.g., ONOO¯, O_2_^•−^) [189], represent central pathomechanisms of neurotoxicity in general [190] and are a hallmark of most neurological complications as well as neuronal/psychiatric diseases in particular [185,188]. Moreover, adverse redox regulation of and by microglia largely contributes to these pathologies [191,192,193]. Microglia recognize carbon black nanoparticles, as contained in cigarette smoke, with the MAC-1 receptor to produce ROS [194] through NOX-2 activation [195]. The different pathophysiological processes initiated by air pollution constituents such as particulate matter that contribute to the neurological and psychiatric health outcomes, as well as the sequence of these events, are summarized in [152]. The very similar processes that account for smoking induced neurotoxicity are shown in Figure 5.

## 5. Conclusions

Despite intense efforts to reduce the global prevalence of smoking and the burden of attributable disease, smoking still remains a leading cause of impaired health and well-being. Importantly, smoking is not only a well-established contributor to global mortality and morbidity by increased risk of non-communicable diseases such as cancer, lung, cardiovascular, and metabolic disease, but may also lead to increased onset of neuropsychiatric disease. It is clear that an interrelationship between smoking and neuropsychiatric disease exists, since smoking is over-represented among individuals with neuropsychiatric disease and vice versa. In this context, a key question remains: whether the effect of smoking on neuropsychiatric disease risk is of causal nature. Indeed, evidence from meta-analyses and large clinical studies is now accumulating to demonstrate that smoking predicts the onset of a wide range of neuropsychiatric disease phenotypes such as dementia/cognitive decline, schizophrenia/psychosis, depression, anxiety disorder, and suicidal behavior. This is of special importance, since neuropsychiatric diseases affect a significant portion of the global population, posing a high burden for individuals and the socioeconomic system. On a mechanistic level, studies indicate primarily oxidative stress, both from free radicals in smoke and from oxidative imbalance in the cells, as well as inflammation involved in the majority of the long-term negative effects of smoking in both neuropsychiatric diseases as well as other smoking-induced non-communicable diseases, thus constituting a central shared pathophysiological pathway. Furthermore, substantial evidence is now available that smoking may lead to changes in brain structure and neuronal function clearly implicated in, and thus predisposing to, neuropsychiatric disorders. Uncertainties and differences in study results may occur due to a variety of reasons that mainly include the more or less reliable assessment of smoking exposure (subjective rather than objective), incomplete adjustment for confounding variables (such as neuropsychiatric comorbidities and psychological distress), and discrepancies in definition of outcome variables (e.g., self-report questionnaire, physician diagnosis, registry data etc.). To this end, it is important to clarify that smoking-related nicotine dependence should not be regarded as a modifiable lifestyle risk factor, but as a manifest mental disorder, with comorbid conditions being common. Of note, adverse childhood experiences were shown to be a crucial risk factor for later nicotine dependence as well as for the development of most mental disorders [196]. From a public health perspective, warnings on tobacco products and other preventive measures to increase the awareness about the detrimental health effects of smoking should necessarily include actions that address the risk of smoking-induced neuropsychiatric disease.

## Figures and Tables

**Figure 1 ijms-22-07272-f001:**
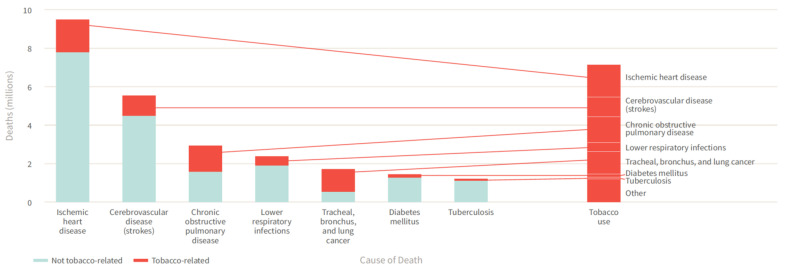
Tobacco use contributes to the leading causes of global deaths including non-communicable diseases as well as infectious diseases. Reused from The Tobacco Atlas, 6th edition with permission. Copyright ©2018 The American Cancer Society, Inc. [3].

**Figure 2 ijms-22-07272-f002:**
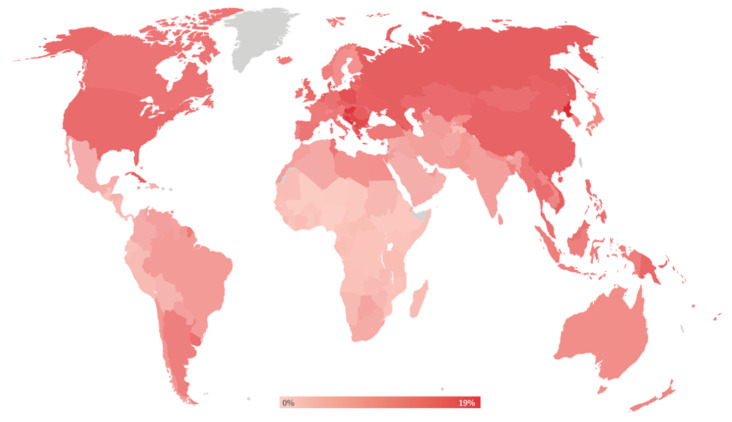
Percent of DALYs attributable to tobacco use. Reused from The Tobacco Atlas, 6th edition with permission. Copyright ©2018 The American Cancer Society, Inc. [3].

**Figure 3 ijms-22-07272-f003:**
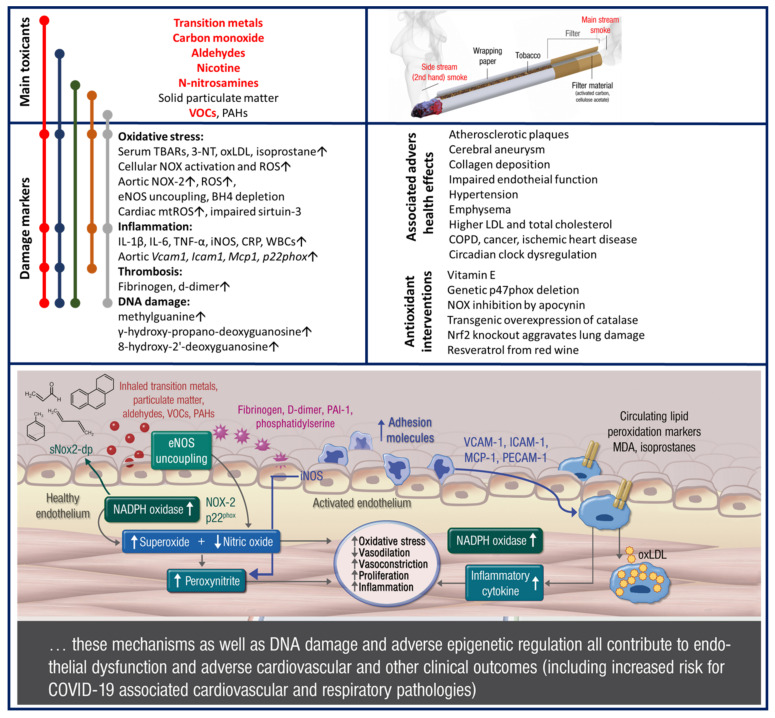
Toxic compounds, pathomechanisms, and adverse health effects of smoking as summarized from human and animal studies: The major toxicants (red and bold = high quantity, black = intermediate quantity) that are associated with smoking are listed along with a schematic description of the specific features of a tobacco cigarette. The molecular link between these toxicants and major damage markers reported for smoking with respect to oxidative stress is shown on the left side. The effects of these toxicants on endothelial (vascular) function are summarized in the inserted scheme. The associated adverse health effects as well as antioxidant interventions are also shown. Modified from Münzel et al. [41] with permission. Copyright © 2020, Oxford University Press.

**Figure 4 ijms-22-07272-f004:**
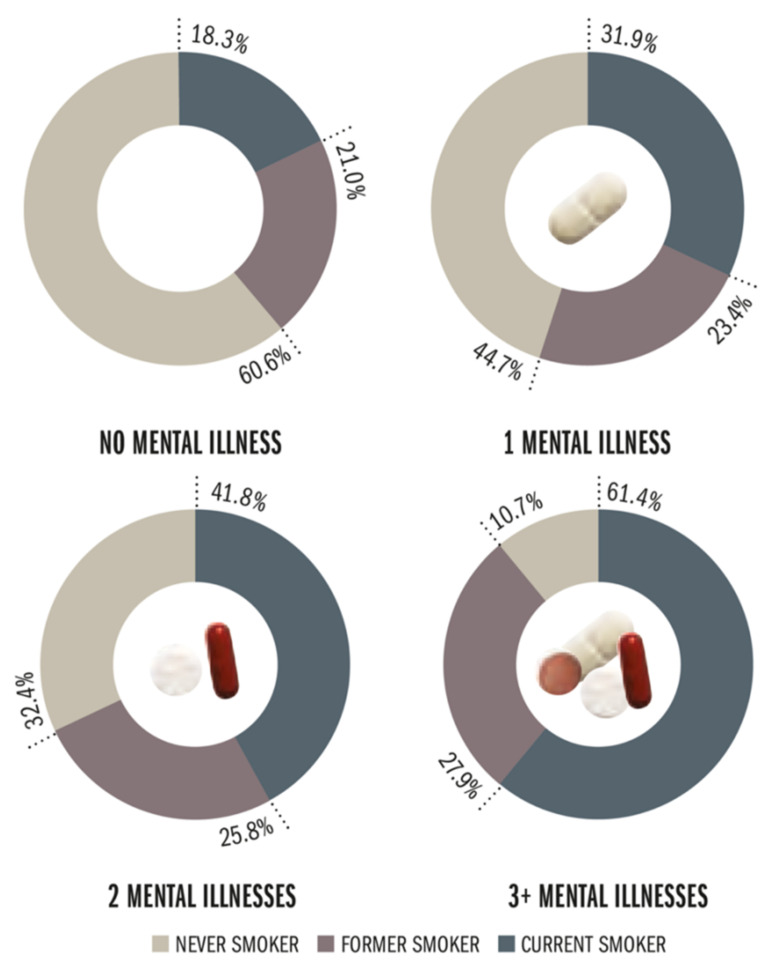
Prevalence of smoking among people with lifetime mental illnesses or psychological distress, comprising bipolar disorder, dementia, schizophrenia, phobias/fears, attention deficit/hyperactivity, and serious psychological distress. Prevalence of current smoking increased with higher numbers of mental illness, ranging from 18.3% for subjects with no illness to 61.4% for subjects with three or more mental illnesses. Reused from The Tobacco Atlas, 5th edition with permission. Copyright ©2015 The American Cancer Society, Inc. [101].

**Figure 5 ijms-22-07272-f005:**
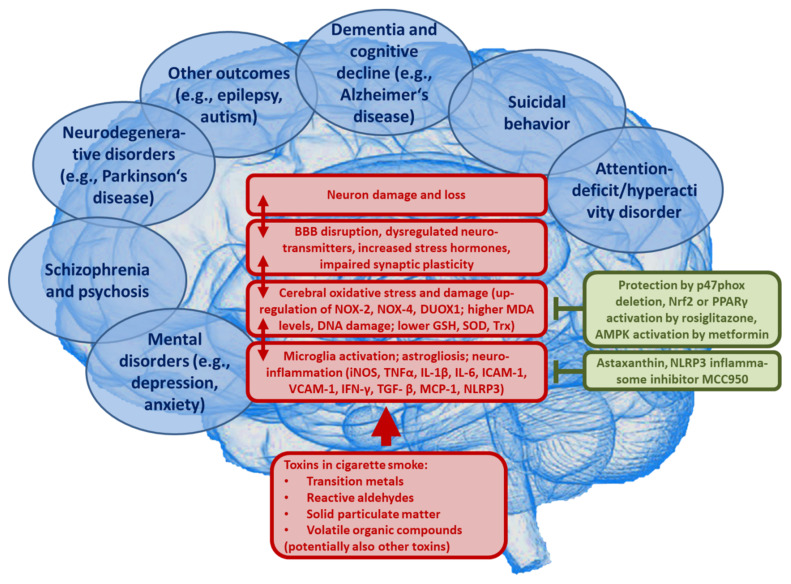
Proposed concept how the major toxins in cigarette smoke may contribute to neurological complications and neuronal/psychiatric diseases. Uptake of fine particulate matter, heavy/transition metals, reactive aldehydes, volatile organic compounds, and other cigarette smoke toxins leads to neuroinflammation and cerebral oxidative stress by microglia activation, impairing vital pathways in the brain, and initiating pathophysiological processes such as amyloid deposition and neuron damage and loss. BBB, blood-brain barrier; iNOS, inducible nitric oxide synthase; TNFα, tumor necrosis factor alpha; IL-1β, interleukin 1beta; IL-6, interleukin 6; NOX-2, NADPH oxidase isoform 2 (phagocytic NADPH oxidase); NOX-4, NADPH oxidase isoform 4; DUOX1, dual oxidase 1; MDA, malondialdehyde; GSH, glutathione; SOD, superoxide dismutase; Trx, thioredoxin; VCAM-1, vascular cell adhesion molecule 1; ICAM-1, intercellular adhesion molecule 1; MCP-1, monocyte chemotactic protein 1 (CCL2); TGF- β, transforming growth factor beta; IFN-γ, interferon gamma; NLRP3, NLR family pyrin domain containing 3; p47phox, regulatory subunit of NOX-2 and NOX-1; Nrf2, nuclear factor E2 related factor-2; PPAR γ, Peroxisome proliferator-activated receptor gamma; AMPK, AMP-activated protein kinase.

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
