# Peer review of "Smoking and Neuropsychiatric Disease—Associations and Underlying Mechanisms"

_ijms, 2021, doi:10.3390/ijms22147272_

Round 1
Reviewer 1 Report
This is a comprehensive and interesting review summarizing current knowledge on the pathological effects of cigarette smoking on different disease contexts, with a focus on neuropsychiatric disorders.
My major criticism is related to three points, being the first two more important and the last more at a suggestion level.
- Neurodegenerative disorders including Alzheimer’s and Parkinson’s deserve a specific section as they are not properly neuropsychiatric disorders although displaying extensive psychiatric symptoms.
- Specific role of nicotinic neuromodulation as a potential mediator of the implication of cigarette smoke in neuropsychiatric disorders has not been discussed.
- Considering the scope of the review, probably the initial part, related to a description of those pathologies with a clear relationship with tobacco smoke, is a bit too discussed and long, but this is just my opinion.
Specific points:
The first sentence of the abstract is hard to read, which does not make a good business card for the paper.
The use of determiners is sometimes unproper.
LINE 27. If the risk itself is elusive, of course underlying mechanisms are not well understood. This sentence is not sharp. I would rephrase.
LINE 69-70. I would substitute “coming from” with “generated by”.
LINE 91. “Accepted that there is a causal link” “officially acknowledged the causal link”
LINE 99. “Indicated that there is at least a correlation” ”Indicated at least a correlation”
LINE 112-114. This sentence is not clear, as it is, it suggests non-dependence of cervical cancer from smoke“indicating that even women who do not smoke are at risk”.
LINE 143. “Is of essence”Is of the essence”
LINE 144. “Is exposed oxidative stress” “Is exposed to oxidative stress”
LINE163. I do not know the verb “caries”
FIG. 3 This referee would not start the written part of the figure with the ellipsis.
LINE 217. Please double check the word “nitrate” Isn’it protein nitrosylation?
LINE 238. Substitute “to variety” with “to a variety”
LINE 264. I wouldn’t say “big role” rather, important… relevant… prominent…
LINE 271. Los must be corrected.
LINE 271 - on. The authors are presenting Alzheimer’s and Parkinson’s diseases as neuropsychiatric or this is a sort of “by the way, also different brain disorders such as neurodegenerative diseases entail oxidative stress as…” or again “psychiatric traits of neurodegenerative disorders could entail oxidative stress as potential pathomechanistic enhancer” I do not understand.
LINE 286. The authors can add a neurodegenerative paragraph between the other smoke-fostered diseases and neuropsychiatric disorders. To treat neurodegenerative disorders within the session of neuropsychiatric ones seem not perfect to me.
LINE 288-292. This notion must be highlighted in such a way that a major issue in this kind of studies is acknowledged.
LINE 298. “Later in life” is a more correct form for this.
LINE 298 on. One of the most convincing and predicting factors linking cigarette smoke with neuropsychiatric disases i.e., nicotine chronic intake and its neuronal effects, deserves a specific paragraph.
LINE 307-309. It is mandatory to underline that psychiatric disorders have a strong link with stress, which actually represents one of the more prominent risk factors for their onset. Well, cancer, cardiovascular and metabolic disorders together with many other chronic pathologies, represent severe forms of chronic stress, intuitively increasing the risk for neuropsychiatric disorders.
LINE 310-312. I do not agree with this conclusion. These concepts, including my former comment must be thoroughly explained.
LINE 364-365. This is the point: stress and anxiety along with depressive drifts may increase tobacco consumption as a strategy to cope with these behavioral aberrations. For sure the anxiolytic effect of cigarette smoke must be discussed. These concepts must be reported with a particular attention to addictive processes. Addiction is per se a psychiatric disorder, the authors should also take this into consideration. The relationship between smoke and neuropsychiatric disorders must always be treated bidirectionally, and the complexity (who was the first?) of the causal links deserve a whole introductory paragraph. The way this section is currently presented is not totally clear.
LINE 405. Association must be corrected.
LINE 448. What does the word “incident” mean? This is not very common and I found it also difficult to understand it surfing the internet.
LINE 504-513. The description of those studies showing smoke protective effects on the incidence of Parkinson disease again seems to fall outside a general description of neuropsychiatric disorders. Indeed, if on the one hand it’s true that Parkinson’s shows extensive traits of anxiety and other psychiatric symptoms, it is classified as neurodegenerative disorder due to loss of dopaminergic neurons at the substantia nigra pars compacta. Neuropsychiatric disorders are curiously gathered also for endemic loss of morphological biomarkers. As well as for Alzheimer’s, this referee would describe the negative and protective effect of cigarette smoke on neurodegenerative disorders in a dedicated section of the review.
LINE 551. Aforementioned is a only word.
LINE 545 and 575. The terminology “neuronal complications” does not convince this reviewer, probably “damage” should be more appropriate, but other terms could ameliorate the significance better than the word “complications”.
LINE 615-618. The link between air pollution and similar components of cigarette smoke is weak, again Alzheimer is not a psychiatric disorder.
LINE 615-621. This part should be included in a specific section Smoke and Neurodegenerative disorders.
LINE 645. Change “Smoking remains still” in “Smoking still remains”.
LINE 652. Change “the effect smoking in “the effect of smoking”
LINE 665. Change “predisposing for” in “predisposing to”
CONCLUSION. This final section acknowledges oxidative stress as a potential leading cause of neuronal dysfunction underlying neuropathology, but the evidences linking smoke with detrimental changes in neuronal function and brain structure are poorly described within the review. If such studies are missing or incomplete it should be mentioned.
How chronic stimulation of nicotine receptors in smokers and animal models change receptor function and plastic neuronal functions in the different brain areas potentially involved in neuropsychiatric drifts including cortical and limbic emotional processing areas? May desensitization of nicotine receptors apply? These receptors are ionotropic channels allowing calcium entry in neurons, and it has been clearly shown that excitatory (and sometimes also inhibitory) neurotransmission dysfunctions play prominent role in the pathogenesis of psychiatric disorders. I think that this review should be enriched by inherent hints.
Author Response
Dear Editor,
Dear Reviewers,
At first, we would like to thank you for taking the time to evaluate our work. We are grateful for the opportunity to improve our manuscript. With regard to the feedback of the reviewer, we would like to respond point by point to the comments as follows. In addition to the clean manuscript version as a word-file, we also provide a version of the revised manuscript with all changes marked to better follow our changes.
Best regards
Andreas Daiber and Thomas Münzel
#Reviewer 1
This is a comprehensive and interesting review summarizing current knowledge on the pathological effects of cigarette smoking on different disease contexts, with a focus on neuropsychiatric disorders.
Response: We thank you for the favorable evaluation of our work.
My major criticism is related to three points, being the first two more important and the last more at a suggestion level.
- Neurodegenerative disorders including Alzheimer’s and Parkinson’s deserve a specific section as they are not properly neuropsychiatric disorders although displaying extensive psychiatric symptoms.
Response: We thank you for this important point. AD was specifically addressed in section 4.1.4 Dementia/Cognitive Decline. PD was specifically addressed in section 4.1.6 Other Neuropsychiatric Outcomes. We referred to neuropsychiatric disease as conditions of psychiatric and/or neurological origin as we did it in a recent review published in the IJMS (https://doi.org/10.3390/ijms21124306).
- Specific role of nicotinic neuromodulation as a potential mediator of the implication of cigarette smoke in neuropsychiatric disorders has not been discussed.
Response: Although this is an interesting point, we feel that nicotinic neuromodulation is beyond the scope of our review as there are >4000 hits in PubMed for “nicotinic neuromodulation smoking”. So this topic definitely deserves a separate detailed review. However, the role of nicotine was addressed in a few points on page 8: “Nicotine, as a highly addictive substance that modifies neurotransmitter patterns, can have a direct influence on neuropsychiatric diseases like schizophrenia and depression. Interestingly, there is also evidence for procognitive effects of nicotine in subjects with neuropsychiatric diseases such as schizophrenia and ADHD, which may explain sustained smoking in these disease phenotypes. In addition, chemicals present in smoke may interact with antipsychotics, antidepressants, and benzodiazepines through pharmacokinetic and pharmacodynamic (mainly nicotine-mediated) pathways, showing that e.g. smoking cessation in patients receiving clozapine may lead to elevated plasma concentrations of clozapine and severe side effects via altered metabolic clearance by CYP1A2.”
In addition, we have now cited two specific reviews on nicotine contribution to neuropsychiatric disorders on page 8: PMID: 19665479, PMID: 10880717, PMID: 1677596
- Considering the scope of the review, probably the initial part, related to a description of those pathologies with a clear relationship with tobacco smoke, is a bit too discussed and long, but this is just my opinion.
Response: Since our aim was to provide an overview of the main disease groups that are clearly linked to smoking, we think that this part is of benefit for the reader. Especially as also cardiovascular and cancer complications by smoking may contribute to neuropsychiatric disorders and share some common pathomechanisms (such as oxidative stress and inflammation) with each other condition.
Specific points:
The first sentence of the abstract is hard to read, which does not make a good business card for the paper.
Response: In line with your suggestion, we reworded this sentence. It reads now: “Despite extensive efforts to combat cigarette smoking/tobacco use, it still remains a leading cause of global morbidity and mortality killing more than eight million people each year.”
The use of determiners is sometimes unproper.
Response: We corrected this accordingly.
LINE 27. If the risk itself is elusive, of course underlying mechanisms are not well understood. This sentence is not sharp. I would rephrase.
Response: We corrected the sentence accordingly. It reads now: “While tobacco smoking is a major risk factor for non-communicable diseases related to the four main groups – cardiovascular disease, cancer, chronic lung disease, and diabetes – its impact on neuropsychiatric risk is rather elusive.”
LINE 69-70. I would substitute “coming from” with “generated by”.
Response: We corrected this accordingly.
LINE 91. “Accepted that there is a causal link” à “officially acknowledged the causal link”
Response: We corrected this accordingly.
LINE 99. “Indicated that there is at least a correlation” à ”Indicated at least a correlation”
Response: We corrected this accordingly.
LINE 112-114. This sentence is not clear, as it is, it suggests non-dependence of cervical cancer from smoke“indicating that even women who do not smoke are at risk”.
Response: We added to word “actively” to clarify that is related to passive smoking. It reads now: “indicating that even women who do not actively smoke are at risk.”
LINE 143. “Is of essence”àIs of the essence”
Response: We corrected this accordingly.
LINE 144. “Is exposed oxidative stress” à “Is exposed to oxidative stress”
Response: We corrected this accordingly.
LINE163. I do not know the verb “caries”
Response: We corrected this accordingly to “carries”.
FIG. 3 This referee would not start the written part of the figure with the ellipsis.
Response: Whereas the upper part of this table/figure combination could be freely modified, the lower part with the scheme and written text could not be modified as it was reproduced from Münzel et al. [41] with permission. Copyright © 2020, Oxford University Press. Changes were not covered by the obtained permission.
LINE 217. Please double check the word “nitrate” Isn’it protein nitrosylation?
Response: ”Nitrate” is correct as nitration describes the formation of a “nitro”-group (R-NO2), e.g. by peroxynitrite in tyrosine residues. In contrast, “nitrosylate” or “nitrosate” would be the transfer of a “nitroso”-group (R-NO), e.g. by N2O3 or NO+ to thiols of cysteine residues.
LINE 238. Substitute “to variety” with “to a variety”
Response: We corrected this accordingly.
LINE 264. I wouldn’t say “big role” rather, important… relevant… prominent…
Response: It now reads: “prominent role”.
LINE 271. Los must be corrected.
It now reads: “Loss of…”.
LINE 271 - on. The authors are presenting Alzheimer’s and Parkinson’s diseases as neuropsychiatric or this is a sort of “by the way, also different brain disorders such as neurodegenerative diseases entail oxidative stress as…” or again “psychiatric traits of neurodegenerative disorders could entail oxidative stress as potential pathomechanistic enhancer” I do not understand.
Response: The aim of this section was to give a first overview of neuropsychiatric diseases (i.e. psychiatric and neurological disorders as explained in the response to your first major comment) that are linked to oxidative stress and inflammation. Thus, we think that in this context the presentation is adequate. However, we changed the sentence on page 7 bottom to make our intention clearer.
LINE 286. The authors can add a neurodegenerative paragraph between the other smoke-fostered diseases and neuropsychiatric disorders. To treat neurodegenerative disorders within the session of neuropsychiatric ones seem not perfect to me.
Response: Please see the response to your first major comment.
LINE 288-292. This notion must be highlighted in such a way that a major issue in this kind of studies is acknowledged.
Response: We added accordingly: “The clear distinction between cause and effect is a major issue in previous studies.”
LINE 298. “Later in life” is a more correct form for this.
Response: We corrected this accordingly.
LINE 298 on. One of the most convincing and predicting factors linking cigarette smoke with neuropsychiatric disases i.e., nicotine chronic intake and its neuronal effects, deserves a specific paragraph.
Response: We totally agree with the importance of nicotine for neuropsychiatric disoders, but we feel, as mentioned above, that this is beyond the scope of the review. As answered above, we cited 3 specific reviews on this topic.
LINE 307-309. It is mandatory to underline that psychiatric disorders have a strong link with stress, which actually represents one of the more prominent risk factors for their onset. Well, cancer, cardiovascular and metabolic disorders together with many other chronic pathologies, represent severe forms of chronic stress, intuitively increasing the risk for neuropsychiatric disorders.
Response: We absolutely agree with reviewer and added on page 8, bottom: “Lastly, it is highly important to note that (neuro)psychiatric disorders have a strong link with chronic stress, which represents one of the more prominent risk factors for their onset. In this context, cancer, cardiovascular and metabolic disorders together with many other chronic pathologies, represent severe forms of chronic stress, intuitively increasing the risk for neuropsychiatric disorders.”
LINE 310-312. I do not agree with this conclusion. These concepts, including my former comment must be thoroughly explained.
Response: We replaced the lines with the conclusion in the previous comment.
LINE 364-365. This is the point: stress and anxiety along with depressive drifts may increase tobacco consumption as a strategy to cope with these behavioral aberrations. For sure the anxiolytic effect of cigarette smoke must be discussed. These concepts must be reported with a particular attention to addictive processes. Addiction is per se a psychiatric disorder, the authors should also take this into consideration. The relationship between smoke and neuropsychiatric disorders must always be treated bidirectionally, and the complexity (who was the first?) of the causal links deserve a whole introductory paragraph. The way this section is currently presented is not totally clear.
Response: Thank you for this important point. We tried to address these points with the following sentences:
“The relationship between smoking, oxidative stress, inflammation, and neuropsychiatric disease is not always clear. This stems from the fact that many neuropsychiatric diseases also increase the chance that a person will start smoking making the direction of association difficult (Figure 4). The clear distinction between cause and effect is a major issue in previous studies.”
“It is clear that an interrelationship between smoking and neuropsychiatric disease exists since smoking is over represented among individuals with neuropsychiatric disease and vice versa. In this context, a key question remains whether the effect of smoking on neuropsychiatric disease risk is of causal nature. Indeed, evidence from meta-analyses and large clinical studies is now accumulating to demonstrate that smoking predicts the onset of a wide range of neuropsychiatric disease phenotypes such as dementia/cognitive decline, schizophrenia/psychosis, depression, anxiety disorder, and suicidal behavior.”
Furthermore, we introduced the section with the following statement in order to highlight that we only focus on studies that allow an assessment of the direction of effects that goes from smoking to incident/new onset of diseases to avoid the question of “who was the first”. “Although there is ample evidence indicating an interrelationship between smoking and neuropsychiatric diseases meaning that rates of smoking are markedly higher in subjects with prevalent neuropsychiatric disease than in the general population being two to five times higher including subjects with e.g. schizophrenia, depression, anxiety disorders, ADHD, binge eating disorder, bulimia, and substance use disorders (Figure 4), much less is known about the prospective impact of smoking on neuropsychiatric disease development. However, emerging strong evidence from epidemiological studies suggests that smoking may be a causal factor for the development or progression of neuropsychiatric disease.”
In addition, we included studies that tested for both directions for the relationship of smoking and neuropsychiatric disease. For example: “In good agreement, in the Korea Welfare Panel smoking was shown to predict depression, whereas no association was found when testing for the opposite direction for the relationship of depression and smoking.”
The point that addiction/nicotine dependence is per se a psychiatric condition was emphasized: “To this end, it is important to clarify that smoking-related nicotine dependence should not be regarded as a modifiable lifestyle risk factor, but as a manifest mental disorder with comorbid conditions being common.”
Since the focus of the review was to highlight studies that reported on the direction from smoking to neuropsychiatric disease, we did not describe the anxiolytic effect of cigarette smoke.
Thus, we think that the organization of these parts fits the scope of the review.
LINE 405. Association must be corrected.
Response: We corrected this accordingly.
LINE 448. What does the word “incident” mean? This is not very common and I found it also difficult to understand it surfing the internet.
Response: “Incident” can be translated to “new onset” and is commonly used when describing the onset of conditions.
LINE 504-513. The description of those studies showing smoke protective effects on the incidence of Parkinson disease again seems to fall outside a general description of neuropsychiatric disorders. Indeed, if on the one hand it’s true that Parkinson’s shows extensive traits of anxiety and other psychiatric symptoms, it is classified as neurodegenerative disorder due to loss of dopaminergic neurons at the substantia nigra pars compacta. Neuropsychiatric disorders are curiously gathered also for endemic loss of morphological biomarkers. As well as for Alzheimer’s, this referee would describe the negative and protective effect of cigarette smoke on neurodegenerative disorders in a dedicated section of the review.
Response: Thank you for this interesting point. Please see the comments above concerning the subsumption of psychiatric and neurological diseases as neuropsychiatric diseases. Although the negative and protective effects of cigarette smoking on neurodegenerative disorders is of special interest, this is beyond the scope of our review.
LINE 551. Aforementioned is a only word.
Response: We corrected this accordingly.
LINE 545 and 575. The terminology “neuronal complications” does not convince this reviewer, probably “damage” should be more appropriate, but other terms could ameliorate the significance better than the word “complications”.
Response: We corrected this accordingly.
LINE 615-618. The link between air pollution and similar components of cigarette smoke is weak, again Alzheimer is not a psychiatric disorder.
Response: Current research shows that the health risks from cigarette smoking and air pollution are similar, thus we think that a parallel can be drawn.
LINE 615-621. This part should be included in a specific section Smoke and Neurodegenerative disorders.
Response: Please see the comments above concerning the subsumption of psychiatric and neurological diseases as neuropsychiatric diseases.
LINE 645. Change “Smoking remains still” in “Smoking still remains”.
Response: We corrected this accordingly.
LINE 652. Change “the effect smoking in “the effect of smoking”
Response: We corrected this accordingly.
LINE 665. Change “predisposing for” in “predisposing to”
Response: We corrected this accordingly.
CONCLUSION. This final section acknowledges oxidative stress as a potential leading cause of neuronal dysfunction underlying neuropathology, but the evidences linking smoke with detrimental changes in neuronal function and brain structure are poorly described within the review. If such studies are missing or incomplete it should be mentioned.
How chronic stimulation of nicotine receptors in smokers and animal models change receptor function and plastic neuronal functions in the different brain areas potentially involved in neuropsychiatric drifts including cortical and limbic emotional processing areas? May desensitization of nicotine receptors apply? These receptors are ionotropic channels allowing calcium entry in neurons, and it has been clearly shown that excitatory (and sometimes also inhibitory) neurotransmission dysfunctions play prominent role in the pathogenesis of psychiatric disorders. I think that this review should be enriched by inherent hints.
Response: Section “4.2 Smoking and Brain Changes Affecting Neuropsychiatric Pathophysiology” including figure 5 aimed at giving a compact overview of the role of oxidative stress as well as inflammation in neuronal dysfunction and underling neuropathology. Although these are interesting points, we feel that our explanations provide an accessible overview for the reader that fits with the main messages.
Reviewer 2 Report
The paper by Hahad and colleagues explores and supports the importance of smoking as a potential risk factor in development of different neuropsychiatric diseases. The authors present the results of different epidemiological and experimental studies that support the association between smoking and neuropsychiatric diseases, and the also discuss pathophysiological mechanisms that may contribute to this relationship.
The paper is well written and it needs only some minor language corrections. The authors should pay attention to the use of commas. Namely, many sentences are difficult to follow precisely because of the lack of commas that should serve to separate certain parts of the sentences. I have only few minor comments or advices for ameliorating the paper and I would suggest that a native English speaker or language professional review the manuscript prior to resubmitting the article.
I would suggest rephrasing some of the sentences to make them easier to follow. For example:
Line 118-119: “Tobacco smoking is also found in literature to be ASSOCIATED with oral cancers [27], head and neck cancers [28], and ovarian cancer [29].” I would suggest deleting “in literature”.
Line 143-144: “When the endothelium is exposed TO oxidative stress or inflammatory me…”
Line 157-158: “It would follow that prolonged exposure to nicotine COULD induce hypertension, but the causal connection is unclear.”
Line 274-275: “Alzheimer’s disease is especially tightly related to oxidative stress as accumulation of metals like iron in neurons produces large amounts of ROS [82].” Please rephrase this sentence.
Line 377-380: “Khaled et al. analyzed data from the Canadian National Population Health Survey to demonstrate that heavy smoking was associated with onset of major depression (HR 3.1, 95% CI 1.9-5.2) even after adjustment FOR mental stress [115].”
Author Response
Dear Editor,
Dear Reviewers,
At first, we would like to thank you for taking the time to evaluate our work. We are grateful for the opportunity to improve our manuscript. With regard to the feedback of the reviewer, we would like to respond point by point to the comments as follows. In addition to the clean manuscript version as a word-file, we also provide a version of the revised manuscript with all changes marked to better follow our changes.
Best regards
Andreas Daiber and Thomas Münzel
#Reviewer 2
The paper by Hahad and colleagues explores and supports the importance of smoking as a potential risk factor in development of different neuropsychiatric diseases. The authors present the results of different epidemiological and experimental studies that support the association between smoking and neuropsychiatric diseases, and the also discuss pathophysiological mechanisms that may contribute to this relationship.
The paper is well written and it needs only some minor language corrections. The authors should pay attention to the use of commas. Namely, many sentences are difficult to follow precisely because of the lack of commas that should serve to separate certain parts of the sentences. I have only few minor comments or advices for ameliorating the paper and I would suggest that a native English speaker or language professional review the manuscript prior to resubmitting the article.
Response: We thank you for the favorable evaluation of our work. In line with your suggestion, we corrected the manuscript according to punctuation, grammar and language.
I would suggest rephrasing some of the sentences to make them easier to follow. For example:
Line 118-119: “Tobacco smoking is also found in literature to be ASSOCIATED with oral cancers [27], head and neck cancers [28], and ovarian cancer [29].” I would suggest deleting “in literature”.
Response: We corrected this accordingly.
Line 143-144: “When the endothelium is exposed TO oxidative stress or inflammatory me…”
Response: We corrected this accordingly.
Line 157-158: “It would follow that prolonged exposure to nicotine COULD induce hypertension, but the causal connection is unclear.”
Response: We corrected this accordingly.
Line 274-275: “Alzheimer’s disease is especially tightly related to oxidative stress as accumulation of metals like iron in neurons produces large amounts of ROS [82].” Please rephrase this sentence.
Response: We corrected this accordingly. It reads now: “Compelling data demonstrate that impaired neuronal metal homeostasis could be involved in formation of oxidative stress influencing amyloid aggregation in case of Alzheimer’s disease.”
Line 377-380: “Khaled et al. analyzed data from the Canadian National Population Health Survey to demonstrate that heavy smoking was associated with onset of major depression (HR 3.1, 95% CI 1.9-5.2) even after adjustment FOR mental stress [115].”
Response: We corrected this accordingly.
Round 2
Reviewer 1 Report
Although I do not fully agree with the authors responses to my criticism, mainly related to the idea that one of the most relevant molecular drive leading from cigarette smoke to behavioral drift is related to neuronal responses to chronic nicotine-mediated stimulation of nicotinic acetylcholine receptors, fundamental for both central and autonomous nervous system function, I decided to valorize the general message of the review, such as the existence of important links between psychopathology and sigarette smoke.
Still I defend the idea that the first part is too long being the title focused on "Smoking and neuropsychiatric disease"
Still I do think that neuronal responses to chronic nicotine are central as pathomechanisms of smoke-related mental disorders.
My position is manly related to the sensation that literature analyzing the negative effects of smoking is important, as smoking epidemics still represents one of the most relevant issue for heath care systems and society.
I think that now, the authors, know my concerns, as such they will be able to decide whether to ameliorate or not these issues.